# Lymphopenia as a Biological Predictor of Outcomes in COVID-19 Patients: A Nationwide Cohort Study

**DOI:** 10.3390/cancers13030471

**Published:** 2021-01-26

**Authors:** Jongmin Lee, Sung-Soo Park, Tong Yoon Kim, Dong-Gun Lee, Dong-Wook Kim

**Affiliations:** 1Division of Pulmonary, Allergy and Critical Care Medicine, Department of Internal Medicine, The Catholic University of Korea, Seoul 06591, Korea; dibs01@catholic.ac.kr; 2Catholic Hematology Hospital, College of Medicine, The Catholic University of Korea, Seoul 06591, Korea; sspark@catholic.ac.kr (S.-S.P.); TYK@catholic.ac.kr (T.Y.K.); 3Division of Hematology, Department of Internal Medicine, College of Medicine, The Catholic University of Korea, Seoul 06591, Korea; 4Division of Infectious Diseases, Department of Internal Medicine, College of Medicine, The Catholic University of Korea, Seoul 06591, Korea

**Keywords:** COVID-19, SARS-CoV-2, mortality, prediction, lymphopenia

## Abstract

**Simple Summary:**

The coronavirus disease 2019 (COVID-19) pandemic caused by the severe acute respiratory syndrome coronavirus 2 (SARS-CoV-2) has caused a major global health crisis. Owing to the rising number of cases and limited global resources, being able to predict patients with a severe disease course is crucial for the initial allocation of the limited medical resources. This study aimed to identify whether lymphopenia is a reliable prognostic marker for COVID-19 using Korean nationwide cohort. Lymphopenia and its severity levels may serve as reliable predictive factors for COVID-19 clinical outcomes including mortality, needs for intensive care, and oxygen requirements. Current study suggests that lymphopenia at the initial presentation of COVID-19 is associated with poor prognosis.

**Abstract:**

We aimed to identify whether lymphopenia is a reliable prognostic marker for COVID-19. Using data derived from a Korean nationwide longitudinal cohort of 5628 COVID-19 patients, we identified propensity-matched cohorts (*n* = 770) with group I of severe lymphopenia (absolute lymphocyte counts [ALC]: <500/mm^3^, *n* = 110), group II of mild-to-moderate lymphopenia (ALC: ≥500–<1000/mm^3^, *n* = 330), and group III, no lymphopenia (ALC: ≥1000/mm^3^, *n* = 330). A significantly higher mortality rate was associated with lymphopenia severity: 40% in group I, 22.7% in group II, and 13.0% in group III (*p* < 0.001). At 28 days, the estimated inferior overall survival associated with intensified lymphopenia: 62.7% in group I, 79.9% in group II, and 89.0% in group III (*p* < 0.001). Lymphopenia contributed significantly toward a greater need for interventions in all groups but at varying degrees: requirements of invasive ventilation, intensive oxygen supply, or adequate oxygen supply, respectively (*p* < 0.001). The lymphopenia intensity was independently associated with higher COVID-19 mortality in multivariable analysis; adjusted odds ratios of 5.63 (95% CI, 3.0–10.72), and 2.47 (95% CI, 1.5–4.13) for group I and group II, respectively. Lymphopenia and its severity levels may serve as reliable predictive factors for COVID-19 clinical outcomes; thus, lymphopenia may provide the prognostic granularity required for clinical use in the management of patients with COVID-19.

## 1. Introduction

In December 2019, a third novel coronavirus emerged in Wuhan, China [1]. A previous study reported that severe acute respiratory syndrome coronavirus 2 (SARS-CoV-2) relates to bat-derived SARS-like coronaviruses [2]. The disease, now called coronavirus disease 2019 (COVID-19), has spread aggressively worldwide. As of 6 November 2020, 48,534,508 confirmed, and 1,231,017 mortality cases were reported to the World Health Organization [3]. Although reports indicate that 80% of patients with SARS-CoV-2 present with a self-limiting manifestation of mild respiratory illness, the 20% minority require hospitalization due to life-threatening organ failure conditions such as acute respiratory distress syndrome, which has led to high mortality [1,4]. Concerning the overwhelming needs of healthcare infrastructure in a pandemic, it is evident that the classification of risk-adapted prognostic factors for patients with COVID-19 could be critical in providing proper allocation of medical resources [5,6]. Accordingly, previous studies aimed to identify prognostic factors of COVID-19 and reported that older age, the presence of comorbidities, and various abnormalities of laboratory test results were reliable parameters related to a fatal prognosis in the hospitalization of patients with COVID-19 [1,7,8]. Nevertheless, the unmet need of novel parameters to optimize risk stratification remains.

Coronaviruses refer to a family of enveloped, positive-sense, single-stranded, and highly diverse RNA viruses [9]. In the last two decades prior to the COVID-19 pandemic, there emerged two highly deadly human pathogenic coronaviruses triggering severe acute respiratory syndrome (SARS-CoV) and the Middle East respiratory syndrome (MERS-CoV) [10]. With the well-known essential role of natural killer cells and cytotoxic T cells in the control of the viral infection [11], previous reports regarding SARS-CoV and MERS-CoV suggested that there could be a reliable correlation between marked lymphopenia (lymphocyte count < 1000/mm^3^) and the severity of the disease [12,13]. In line with the suggestive impact of lymphopenia on SARS-CoV and MERS-CoV, recent COVID-19 investigators, have also focused on the clinical significance of lymphopenia as a prognostic marker for patients with COVID-19 [14,15,16,17,18,19,20,21,22]; nevertheless, the publications, thus far, had limitations relative to the retrospective design nature, small sample sizes, and the single or limited number of institutional involvement. Moreover, these studies did not clearly reveal the impacts of lymphopenia on the clinical outcomes due to the lack of models that could assess possible covariables. Furthermore, binary classification of lymphopenia was restrictively used in either study, although lymphopenia could be classified at various disease intensity levels [23].

To fill in these gaps, we conducted a nationwide longitudinal cohort study with COVID-19 patients in South Korea (hereafter “Korea”) whose data included detailed clinical characteristics, medical history, complete blood counts with lymphocyte counts, the severity of the disease course, and hospitalization periods. We first compared the clinical outcomes of COVID-19 in patients with lymphopenia (absolute lymphocyte counts [ALCs] < 1000/mm^3^) to those without lymphopenia (ALCs: ≥1000/mm^3^) [15], using the total cohort (*n* = 4052), which consisted of the entire study population with available baseline ALC data. Beyond these explorations in the total cohort, we conducted propensity-matched analysis to evaluate the clinical outcomes depending on the degree of lymphopenia with the intention to balance other baseline variables. Lastly, we performed sub-group analysis of patients harboring characteristics that were noted to have a potential relationship to baseline lymphopenia in a total cohort analysis.

## 2. Results

### 2.1. Characteristics of the Total Cohort

Among 5628 patients with COVID-19 registered in the database, we first excluded seven patients diagnosed with COVID-19 after death, 27 patients with no COVID-19 course data, and 1542 patients without baseline data of complete blood counts. For the total cohort consists of 4052 patients having baseline ALC data, we investigated those whose factors associated with lymphopenia. As shown in Table 1, univariate analysis revealed that 19 variables were potentially associated with lymphopenia with a *p* < 0.05. Among these 19 variables, 10 variables, including age (≥60 years), male, diastolic blood pressure < 80 mmHg, body temperature (≥38 °C), dyspnea on presentation, comorbidities of hypertension, renal disease, and dementia, anemia (hemoglobin < 12.5 g/dL), and thrombocytopenia (platelet < 100,000/mm^3^), remained as independent predictors in multivariable analysis. A log-rank test showed that the 28-day overall survival (OS) rate for patients with lymphopenia (82.6%; 95%CI, 79.8–85.6; *n* = 786) was significantly poorer than that of patients without lymphopenia (98.0%; 95% CI, 97.4–98.5; *n* = 3266; *p* < 0.001) (Figure 1).

### 2.2. Baseline Characteristics among Propensity Matched Cohorts

As shown in Figure 1, the 4052 patients of the total cohort were classified into three groups according to their absolute lymphocyte counts (ALCs) at admission: (I) patients with severe lymphopenia (ALC < 500/mm^3^): *n* = 110, (II) patients with mild to moderate lymphopenia (ALC, ≥500–<1000/mm^3^): *n* = 676, and (III) patients with no lymphopenia (ALC ≥ 1000/mm^3^, *n* = 3266). Thereafter, based on 110 patients with severe lymphopenia (named “group I” hereafter), we established two propensity sub-cohorts from the 676 patients with mild to moderate lymphopenia in group II and from the 3266 patients with no lymphopenia in group III. Group II (*n* = 330) and group III (*n* = 330) were each matched 1:3 to group I (*n* = 110) based on the propensity score (Figure 2). Relative to the propensity score matching, the nearest neighbor matching method was used to match patients in group II and group III with the closest propensity score for each patient in group I. Variables including age, sex, systolic and diastolic blood pressure, baseline heart rate and body temperature, presentations at admission (sputum, fatigue, dyspnea, mental disturbance, nausea/vomiting, and diarrhea), medical comorbidities (on active treatment for cancer, diabetes, hypertension, chronic cardiac disease, chronic renal disease, chronic hepatic disease, autoimmune disease, and dementia), and baseline blood cell counts (hemoglobin, whole blood counts, and platelet) were used as a covariate for matching.

Details of the patient’s demographic and clinical characteristics are listed in Table 2. Of the 770 patients in the total cohort, 66.9% (*n* = 515) of patients were ≥60 years old. The mean ALC of the total cohort was 1088.7 ± 694.0. According to the three groups definition, the lowest mean ALCs was assigned to group I (376.0 ± 106.4/mm^3^), followed by medium-level ALCs of group II (787.3 ± 136.7/mm^3^) and normal level ALCs of group III (1627.6 ± 742.5/mm^3^), determined to be statistically significant (*p* < 0.001). There was a higher trend of platelet counts in group III (a mean of 199,632 ± 74,163/mm^3^), compared with those in group I (a mean of 184,946 ± 98,719/mm^3^) and group II (186,860 ± 70,172/mm^3^) (*p* = 0.058), the *post-hoc* analysis, using the statistical significance of *p* < 0.017, revealed no significant intergroup differences: *p* = 0.824, for the comparison between group I and group II; *p* = 0.1, for a comparison between group I and group III; *p* = 0.026 for a comparison between group II and group III. Other characteristics were well balanced among the three groups. 

### 2.3. Primary and Secondary Endpoints

Of the total cohort (*n* = 770), 21% of patients (*n* = 162) died of COVID-19 during hospitalization. The mortality rate was significantly higher in group I (44 of death patients, 40%), followed by group II (75 of death patients, 22.7%), and group III (45 of death patients, 13.0%) (*p* < 0.001). Kaplan-Meier analyses also revealed inferior overall survival in the group with more severe lymphopenia (*p* < 0.001), with 28-day overall survival rates of group I as follows: 62.7% (95% CI, 54.0–72.9%), group II: 79.9% (95% CI, 75.4–84.7), and group III: 89.0% (95% CI, 85.6–92.5) (Figure 3). Furthermore, we observed similar and significant trends in secondary endpoint measurements such as requirements of invasive ventilation (40.9% in group I, 24.5% in group II, and 14.8% in group III, *p* < 0.001), intensive oxygen supplements (47.3% in group I, 31.8% in group II, and 18.2% in group III, *p* < 0.001), and mild oxygen supplements (72.7% in group I, 55.5% in group II, and 34.2% in group III, *p* < 0.001) as the primary endpoint (mortality rate). Survival periods for patients who died during admission were shortest in group I: 23.4 ± 13.9 days, followed by group II: 25.5 ± 13.4 days, and least with group III: 15.6 ± 14.4, whereas the hospitalized period of alive patients in group I, was longer (29.4 ± 10.8 days) than those in group II (28.2 ± 12.1 days) or in group III (28.0 ± 10.5 days), despite the non-statistical significance (Table 3). The primary and secondary endpoint data are summarized in Table 3.

### 2.4. Prognostic Parameters for Mortality of COVID-19 

We used 16 potential variables for predicting mortality of COVID-19 derived from univariate analysis (Appendix A) to perform multivariate analysis to identify significant parameters associated with mortality of COVID-19. We found that as lymphopenia became more severe, the more it significantly impacted the association with COVID-19 mortality [adjusted odds ratios of 2.47 (95% CI, 1.5–4.13) and 5.63 (95% CI, 3.0–10.72) for group II and group I, respectively], even after adjustment for other potential factors (Table 4). Although diarrhea at presentation, hypertension, chronic heart disease, and chronic pulmonary disease were related to morality in univariable analysis, multivariable analysis revealed that they were not statistically significant. 

Among identified 12 significant parameters in multivariable analysis, the highest odds ratio is shown in mental disturbance (11.09 of adjusted odd ratio (95% CI, 3.28–47.25)), followed by dementia (6.55 of adjusted odd ratio (95% CI, 3.84–11.4)), group I, age ≥ 60 years (4.19 of adjusted odd ratio (95% CI, 2.14–8.82)), dyspnea at presentation (4.18 of adjusted odd ratio (95% CI, 2.61–6.81)), chronic renal disease (3.36 of adjusted odd ratio (95% CI, 1.32–8.54)), treating cancer (3.15 of adjusted odd ratio, (95% CI, 1.43–6.85)), and others (Figure 4). 

### 2.5. Subgroup Analysis

Regarding the associations between baseline lymphopenia and the specific 10 variables previously identified in the analysis of the total cohort (Table 1), i.e., age (≥60 years), male sex, diastolic blood pressure < 80 mmHg, body temperature (≥38 °C), dyspnea on presentation, comorbidities of hypertension, renal disease, dementia, anemia (hemoglobin < 12.5 g/dL), and thrombocytopenia (platelet < 100,000/mm^3^), we performed an additional analysis to explore the impact of lymphocytopenia on the survival outcomes in each subgroup of patients who possessed a particular variable. Moreover, we also analyzed the impacts of lymphopenia on the survival outcomes in the subgroup of patients who were on active treatment for cancer or had autoimmune disease. The subgroup analysis showed that except for a subgroup of patients with renal disease or autoimmune disease, the impact of lymphopenia on survival outcomes were reproduced with statistical significances (Figure 5A–K). Despite a lack of statistical significance in the subgroup of patients with renal disease or autoimmune disease due to the small sample size of each subgroup, enhanced lymphopenia contributed to the development of the fatal course of COVID-19 (Figure 5G,L).

## 3. Discussion

In this nationwide study, we evaluated the impacts of lymphopenia on clinical outcomes of patients with COVID-19. The mean ALC level at admission was inversely associated with COVID-19 mortality in this study, and this association remained significant even after adjusting for confounding factors. Moreover, an enhanced degree of lymphopenia was significantly correlated with the severity of the disease: Each mortality or degree of COVID-19 severity was associated independently with the grades of lymphopenia (severe lymphopenia, ALC < 500/mm^3^; moderate lymphopenia, ALC ≥500–<1000/mm^3^; no lymphopenia, ALC ≥ 1000/mm^3^). As far as we know, this was a novel finding that has not been discussed in previous literature. We added strength to our investigation by conducting a nationwide cohort study, analyzing the largest sample size, compared to previous studies that seemed to have focused on lymphopenia as a prognostic factor for COVID-19. We believe that due to the large-sized sample of this study, our findings will have a higher generalizability. 

The total cohort consisted of 4052 patients with COVID-19 in the current study; 19.4% of patients presenting lymphopenia at admission: 2.7% (*n* = 110) of severe lymphopenia and 16.7% (*n* = 676) of mild to moderate lymphopenia. Previously reported rates of lymphopenia from China ranged from 26 to 80% [24]. In a large-sized cohort study from the United States with available data from of 5645 patients’ lymphocyte count, 60% of patients presented lymphopenia on an initial laboratory test [25]. We believe that this disparity in frequency occurred due to either the different indication for admission treatment such as the extent of participation of the subclinical patient in the cohort, or different definitions for lymphopenia. In Korea, all patients diagnosed with COVID-19 following positive results of PCR tests using nasopharyngeal swab should have been admitted to dedicated facilities, even in the absence of COVID-19-related symptoms [26]. Previous data from two Korean dedicated centers for COVID-19 revealed 17.9% lymphopenia in a total of 352 patients: The proportion of lymphopenia among the patients is comparable to that of our cohort. As far as we know, there have been four cohort studies that explored both the mortality rate of COVID-19 of entire subject in their cohort and a proportion of lymphopenia, as defined by ALC < 1000/mm^3^, identical to that in our study (Appendix A) [17,25,27,28] Interestingly, the order of mortality for each evaluation was consistent with that of a proportion of lymphopenia. These results could illustrate that mild lymphopenia, defined as ALC < 1000/mm^3^, could be a feasible biomarker to predict mortality of COVID-19. Nevertheless, the validation of the clinical feasibility of severe lymphopenia (defined as ALC < 500/mm^3^ in our study), comparing mild-to-moderate lymphopenia or no lymphopenia, in predicting mortality of COVID-19 would be needed in future research.

It is well known that cytotoxic lymphocytes such as cytotoxic T lymphocytes and natural killer cells play an essential role in maintaining immune homeostasis and inflammatory response to control viral infection [14]. Previous studies reported apoptosis or functional exhaustion of cytotoxic lymphocytes associated with the progression of viral infection [29,30]. Although an understanding of the pathogenic mechanism of lymphopenia on the severe course of COVID-19 is still lacking, we were prompted to hypothesize the generation of excessive pro-inflammatory cytokines due to the COVID-19 infection, and expected to progress severely, continue to be disordered, and induce robust lymphocyte apoptosis [31]. Considering that the severe-type COVID-19 disease was associated with elevated blood lactic acid level [32], it is also plausible that inhibition of lymphocyte caused by hyperlactacidemia is linked to severe COVID-19 infection [33]. Our results support a proposal by Bermejo-Martin et al. [34]: novel drugs targeting lymphocyte proliferation or apoptosis (IL-7 or PD-1/PD-L1 inhibitors), that having action of mechanism associated with restore lymphocyte, could be worthy for patients suffering from a severe course of COVID-19. Our study also illustrated that lymphopenia could be an essential parameter if future study has a plan to develop a risk model for COVID-19. Furthermore, for patients with severe lymphopenia, early hospitalization and using currently available treatment options should be considered.

This study had several limitations. First, it was a retrospective cohort study, and selection bias might exist despite using propensity score matching to minimize this effect. Second, the follow-up duration was short; hence, there were no data on post-hospitalization outcomes. Third, although previous studies reported that functional exhaustion of CD8+ and CD4+ T cells is related to the disease severity of COVID-19 [30,35], lymphocyte subsets including B cells, CD4+ and CD8+ T cells, and natural killer cells, were not measured in this study. Fourth, the type and timing of treatments were not included as variables in this study because consensus regarding treatment was not established during patient recruitment. Fifth, we could not confirm the cause of lymphopenia among the subjects; however, this study focused on associations between outcomes of COVID-19 and limited variables such as clinical characteristics, including laboratory findings. Nevertheless, we found that lymphopenia was significantly related to severities and mortality of patients diagnosed with COVID-19, even after adjusting for confounding factors. Additionally, the degree of lymphopenia was significantly associated with oxygen supplement requirements. Moreover, lymphopenia had a convincing impact on adverse survival outcomes even in each subgroup whose key criteria could be closely associated with lymphopenia. Considering the easy procedure involved in complete blood counts and cost-effectiveness, lymphopenia could serve as a prognostic tool for predicting the severity and poor prognosis of COVID-19 in a primary clinic. Therefore, it could be a useful biomarker to consider for risk-adapted medical resource allocation in this pandemic period.

## 4. Materials and Methods 

### 4.1. Data Source

Korea Disease Control & Prevention Agency (KDCA) is a pivotal government agency that controls communicable diseases, including COVID-19. The agency enforced regulation, ensuring that all patients diagnosed with COVID-19 in Korea, regardless of whether they had COVID-19-related symptoms, are admitted to designated hospitals, and must be in quarantine. Clinicians who bear the responsibility of COVID-19 patients have obligations to report the patients’ clinical data with a case report form requested by the KDCA (http://icreat.nih.go.kr) since the patients had been in admission. A case report form includes demographic and epidemiological variables, complete blood counts at admission, and the final treatment course during hospitalization. In July 2020, the KDCA released registered data collected from 5628 patients diagnosed with COVID-19 between January and April 2020. Our current study was approved by the Institutional Review Board of Seoul St. Mary’s Hospital. Written informed consent was waived due to the deidentification and retrospective design of the study (Study approval No. KC20ZADI0654). This study was conducted following the ethical principles established by the Helsinki Declaration update of 2013.

### 4.2. Outcome Measurements

A case of COVID-19 infection was defined, irrespective of clinical signs and symptoms, based on laboratory positive results on SARS-CoV-2 reverse transcription polymerase chain reaction assays by the Korea Ministry of Food and Drug Safety approved kit [36,37]. In deployed data, maximal clinical severity of COVID-19 was categorized into eight levels based on performance of subject, requirements of oxygen, and presence of organ failure as follows [38]: (level I) no limit of activity, (level II) limited activity without oxygen supplementation, (level III) requirement of oxygen supply with nasal cannula, (level IV) requirement of oxygen supply with facial mask, (level V) requirement of non-invasive ventilation, (level VI) requirement of invasive ventilation, (level VII) requirement of extracorporeal membrane oxygenation for multiple organ failure, or (level VIII) death. We defined supportive care, including oxygen supply with invasive ventilation and/or extracorporeal membrane oxygenation, for patients who presented life-threatening organ failure as invasive intensive care.

The primary endpoint was a comparison of mortality rates among the three groups. Secondary endpoints included comparisons of proportions of patients who required invasive intensive care, intensive supplemental oxygen therapy (such as facial mask or nasal cannula oxygen provision, non-invasive positive pressure ventilation, or high flow oxygen administration), mild oxygen supplements via nasal cannula, hospitalized periods for patients who died (survival days), and hospitalized period for alive patients. Additionally, we attempted to identify an association with other baseline parameters, including aforementioned covariates. 

### 4.3. Statistical Analysis

Categorical variables are presented as frequencies and percentages, whereas continuous variables are shown by means ± standard deviations. The three groups’ differences in baseline demographic and clinical characteristics and primary and secondary endpoints were compared using the chi-square test for categorical variables and one-way analysis of variance (ANOVA) for continuous variables. Post hoc analysis with Bonferroni correction for multiple comparisons was performed. The Kaplan–Meier method was used to analyze time-to-event endpoints, such as overall survival, and compared with the log-rank test. A chi-square test was also used to explore associations between endpoint including lymphopenia and death event and clinical variables: variables with *p* < 0.05 in univariable analyses were entered into multivariable models. For multivariable analysis was employed to identify factors associated with presence of lymphopenia using mother cohort and mortality using propensity matched cohorts, respectively, logistic-regression analysis was performed; final parameters associated with mortality were founded as those for which *p* < 0.05 in the multivariable model. We used the R statistical software (ver. 3.6.1, R Foundation for Statistical Computing, Vienna, Austria, 2019) for all statistical analyses. Statistical significance was set at *p* < 0.05.

## 5. Conclusions

There has been no certain treatment for COVID-19 with severe manifestation, but only preventive intervention such as public control of transmission has been introduced as a standard approach for managing COVID-19 [39,40]. Therefore, current result suggested that lymphopenia at the initial presentation of COVID-19 is associated with poor prognosis.

## Figures and Tables

**Figure 1 cancers-13-00471-f001:**
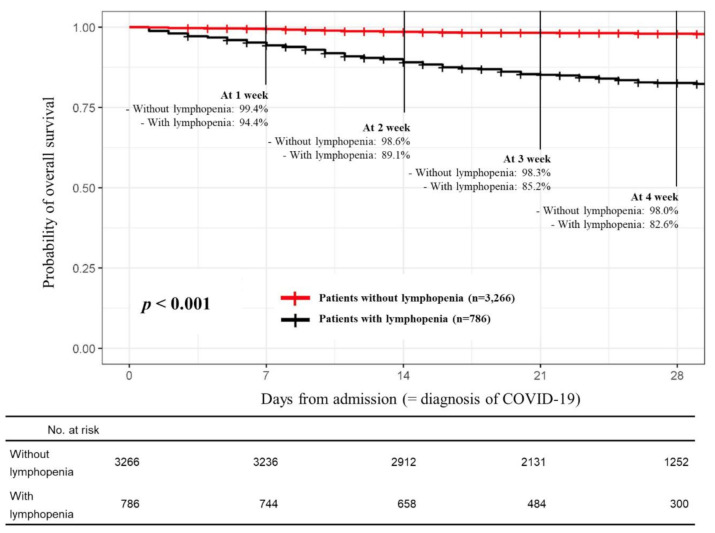
Comparison of the overall survival outcomes of patients with or without lymphopenia in the total cohort.

**Figure 2 cancers-13-00471-f002:**
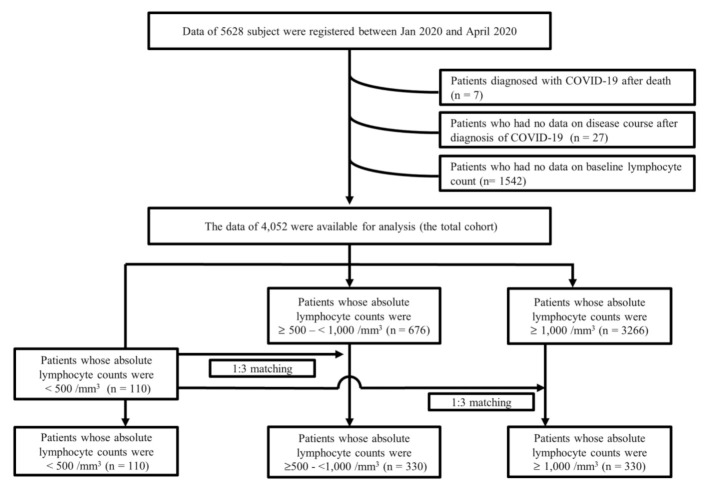
Flow chart showing the disposition of patients according to lymphocyte counts from the Korean nationwide cohort.

**Figure 3 cancers-13-00471-f003:**
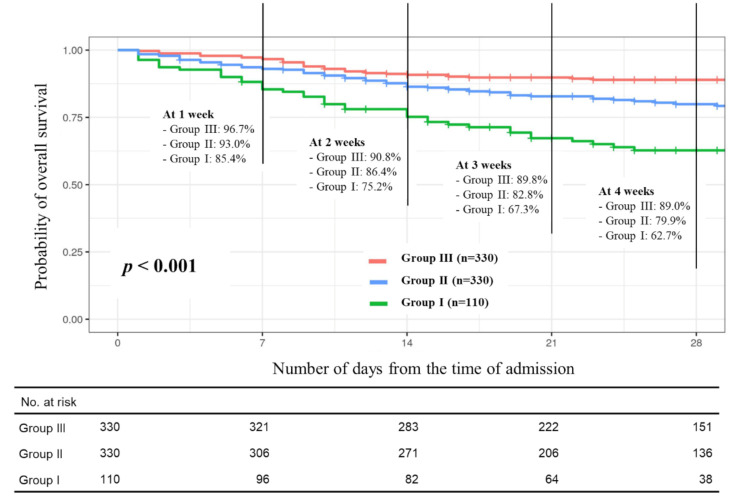
Overall survival outcome according to the lymphopenia groups.

**Figure 4 cancers-13-00471-f004:**
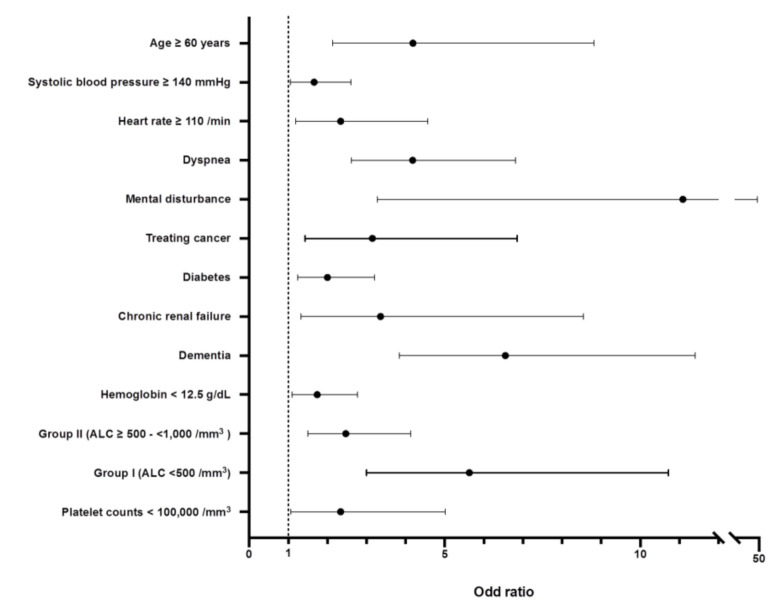
Odds ratios of identified parameters associated with the morality of COVID-19. Abbreviation: ALC, absolute lymphocyte count.

**Figure 5 cancers-13-00471-f005:**
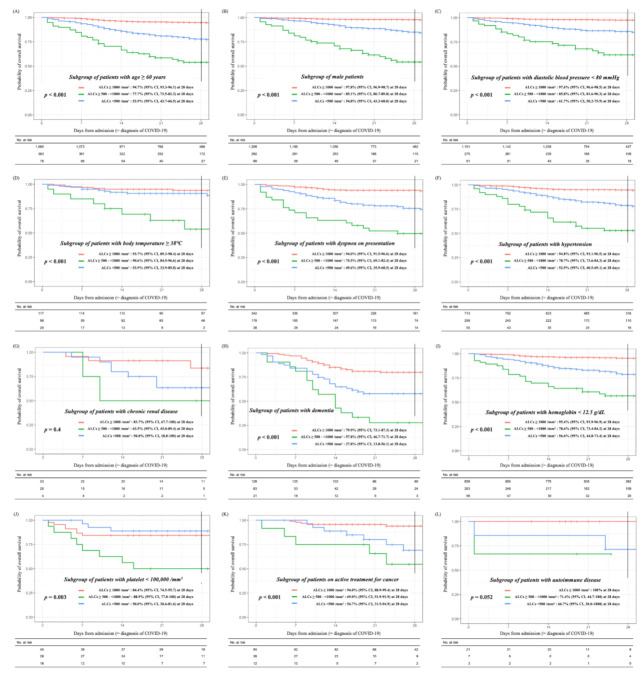
Overall survival outcomes for lymphopenia according to subgroups. The subgroups are as follows: (**A**) patients with age ≥ 60 years (*n* = 1546), (**B**) male patients (*n* = 1570), (**C**) patients with diastolic blood pressure < 80 mmHg (*n* = 1487), (**D**) patients with body temperature ≥ 38 °C (*n* = 235), (**E**) patients with dyspnea on presentation (*n* = 558), (**F**) patients with hypertension (*n* = 1021), (**G**) patients with renal disease (*n* = 47) (**H**) patients with dementia (*n* = 213) (**I**) patients with hemoglobin <12.5 g/dL (*n* = 1178), (**J**) patients with platelet < 100,000/mm^3^ (*n* = 89), (**K**) patients on active treatment for cancer (*n* = 133), and (**L**) patients with autoimmune disease (*n* = 31). Abbreviation: ALC, absolute lymphocyte count.

**Table 1 cancers-13-00471-t001:** Baseline characteristics of total cohort with/without lymphopenia (absolute lymphocyte counts [ALCs] < 1000/mm^3^).

Variables (Total, *n* = 4052)	Univariable Analysis			Multivariable Analysis	
	ALCs < 1000/mm^3^ (*n* = 786)	ALCs ≥ 1000/mm^3^ (*n* = 3266)	*p*-Value	Odds Ratio (95% CI)	*p*-Value
Age			<0.001		<0.001
<60 years	325 (41.3%)	2181 (66.8%)		1	
≥60 years	461 (58.7%)	1085 (33.2%)		1.75 (1.44–2.13)	
Sex			<0.001		<0.001
Female	425 (54.1%)	2057 (63.0%)		1	
Male	361 (45.9%)	1209 (37.0%)		1.56 (1.3–1.87)	
Systolic blood pressure, baseline (missing, *n* = 82)			0.143		-
<140 mmHg, no (%)	474 (61.2%)	2045 (64.1%)		-	
≥140 mmHg, no (%)	301 (38.8%)	1147 (35.9%)		-	
Diastolic blood pressure, baseline (missing, *n* = 175)			<0.001		<0.001
<80 mmHg, no (%)	336 (43.4%)	1151 (36.1%)		1	
≥80 mmHg, no (%)	349 (56.6%)	2041 (63.9%)		0.75 (0.63–0.9)	
Heart rate, baseline (missing, *n* = 71)			0.007		0.393
<110/min	701 (90.2%)	2984 (93.1%)		1	
≥110/min	76 (9.8%)	220 (6.9%)		1.15 (0.83–1.57)	
Body temperature, baseline (missing, *n*= 13)			<0.001		<0.001
<38 °C	666 (84.9%)	3138 (96.4%)		1	
≥8 °C	118 (15.1%)	117 (3.6%)		3.73 (2.77–5.02)	
Presentation of sputum, baseline			0.797		-
Yes	227 (28.9%)	961 (29.4%)		-	
No	559 (71.1%)	2305 (70.6%)		-	
Presentation of fatigue, baseline			<0.001		0.061
Yes	52 (6.6%)	118 (3.6%)		1	
No	734 (93.4%)	3148 (96.4%)		0.69 (0.48–1.02)	
Presentation of dyspnea, baseline			<0.001		<0.001
Yes	216 (27.5%)	342 (10.5%)		1	
No	570 (72.5%)	2924 (89.5%)		0.42 (0.34–0.53)	
Presentation of mental disturbance, baseline			<0.001		0.623
Yes	15 (1.9%)	14 (0.4%)		1	
No	771 (98.1%)	3252 (99.6%)		0.81 (0.35–1.89)	
Presentation of nausea/vomiting, baseline			0.002		0.136
Yes	58 (7.4%)	151 (4.6%)		1	
No	728 (92.6%)	3115 (95.4%)		0.76 (0.54–1.1)	
Presentation of diarrhea, baseline			0.327		-
Yes	81 (10.3%)	297 (9.1%)		-	
No	705 (89.7%)	2969 (90.9%)		-	
Comorbidity of cancer with active treatment ^#^			0.002		0.084
Yes	40 (5.1%)	93 (2.8%)		1	
No	746 (94.9%)	3173 (97.2%)		0.69 (0.45–1.06)	
Comorbidity of diabetes			<0.001		0.381
Yes	181 (23.0%)	421 (12.9%)		1	
No	605 (77.0%)	2845 (87.1%)		0.9 (0.72–1.13)	
Comorbidity of hypertension			<0.001		0.01
Yes	308 (39.2%)	713 (21.8%)		1	
No	478 (60.8%)	2553 (78.2%)		0.76 (0.63–0.94)	
Comorbidity of chronic cardiac disease			<0.001		0.956
Yes	62 (7.9%)	130 (4.0%)		1	
No	724 (92.1%)	3136 (96.0%)		1.01 (0.45–1.45)	
Comorbidity of pulmonary disease ^$^			0.035		0.962
Yes	36 (4.6%)	98 (3.0%)		1	
No	750 (95.4%)	3168 (97.0%)		0.99 (0.65–1.56)	
Comorbidity of chronic renal disease			<0.001		0.019
Yes	24 (3.1%)	23 (0.7%)		1	
No	762 (96.9%)	3243 (98.6%)		0.46 (0.24–0.88)	
Comorbidity of hepatic disease ^†^			0.019		0.485
Yes	21 (2.7%)	46 (1.4%)		1	
No	765 (97.3%)	3220 (98.6%)		0.81 (0.45–1.49)	
Comorbidity of autoimmune disease			0.112		-
Yes	10 (1.3%)	21 (0.6%)		-	
No	776 (98.7%)	3245 (99.4%)		-	
Comorbidity of dementia			<0.001		<0.001
Yes	84 (10.7%)	129 (3.9%)		1	
No	702 (89.3%)	3137 (96.1%)		0.55 (0.4–0.76)	
Hemoglobin (missing, *n* = 7)			<0.001		<0.001
<12.5 g/dL	319 (40.6%)	859 (26.4%)		1	
≥12.5 g/dL	467 (59.4%)	2400 (73.6%)		0.67 (0.55–0.81)	
Platelet counts (missing, *n* = 1)			<0.001		0.001
<100,000/mm^3^	44 (5.6%)	45 (1.4%)		1	
≥100,000/mm^3^	742 (94.4%)	3220 (98.6%)		0.43 (0.26–0.7)	

Abbreviation: SD, standard deviation; ALC, absolute lymphocyte counts ^#^ For a patient whose cancer had been completely cured was not accounted for in the analysis; ^$^ Comorbidities such as asthma or chronic obstructive lung disease were included; ^†^ Subjects with chronic hepatitis were included.

**Table 2 cancers-13-00471-t002:** Baseline demographic and clinical characteristics.

Variables	Absolute Lymphocyte Counts (Total, *n* = 770)			*p*-Value
	Group I, <500/mm^3^ (*n* = 110)	Group II, ≥500–<1000/mm^3^ (*n* = 330)	Group III, ≥1000/mm^3^ (*n* = 330)	
Age				
<40 years, no (%)	7 (25.4%)	25 (7.6%)	28 (8.5%)	0.808
40–59 years, no (%)	25 (25.4%)	89 (27.0%)	81 (24.5%)	
≥60 years, no (%)	78 (70.9%)	216 (65.5%)	221 (67.0%)	
Gender				0.669
Female, no (%)	41 (37.3%)	138 (41.8%)	138 (41.8%)	
Male, no (%)	69 (62.7%)	192 (58.2%)	192 (58.2%)	
Systolic blood pressure, baseline				
<140 mmHg, no (%)	68 (61.8%)	198 (60.0%)	206 (62.4%)	0.475
≥140 mmHg, no (%)	40 (36.4%)	131 (39.7%)	122 (58.2%)	
Missing, no (%)	2 (1.8%)	1 (0.3%)	2 (0.6%)	
Diastolic blood pressure, baseline				
<80 mmHg, no (%)	61 (55.5%)	164 (49.7%)	186 (56.4%)	0.168
≥80 mmHg, no (%)	47 (42.7%)	165 (50.0%)	142 (43.0%)	
Missing, no (%)	2 (1.8%)	1 (0.3%)	2 (0.6%)	
Heart rate, baseline				
<110/min, no (%)	93 (87.5%)	291 (88.2%)	295 (89.1%)	0.443
≥110/min, no (%)	17 (15.5%)	39 (11.8%)	36 (10.9%)	
Missing, no (%)	2 (1.8%)	1 (0.3%)	2 (0.6%)	
Body temperature, baseline, °C, mean ± SD	37.1 ± 0.8	37.1 ± 0.5	37.1 ± 0.8	0.855
<38 °C	90 (81.8%)	275 (83.3%)	294 (89.1%)	0.895
≥38 °C	20 (18.2%)	55 (16.7%)	36 (10.9%)	
Baseline presentation				
Sputum (+), no (%)	38 (34.5%)	105 (31.8%)	109 (33.0%)	0.860
Fatigue (+), no (%)	7 (6.4%)	17 (5.2%)	17 (5.2%)	0.872
Dyspnea (+), no (%)	38 (34.5%)	105 (31.8%)	109 (33.0%)	0.252
Mental disturbance (+), no (%)	3 (2.7%)	9 (2.7%)	8 (2.4%)	0.966
Nausea/vomiting (+), no (%)	9 (8.2%)	22 (6.7%)	28 (8.5%)	0.664
Diarrhea (+), no (%)	16 (4.5%)	46 (13.9%)	52 (15.8%)	0.803
Comorbidity				
Cancer with active treatment ^#^ (+), no (%)	12 (10.9%)	23 (7.0%)	20 (6.1%)	0.860
Diabetes (+), no (%)	31 (28.2%)	88 (26.7%)	90 (27.3%)	0.951
Hypertension (+), no (%)	50 (45.5%)	148 (44.8%)	139 (42.1%)	0.724
Chronic cardiac disease (+), no (%)	12 (10.9%)	32 (9.7%)	34 (10.3%)	0.927
Chronic pulmonary disease ^$^ (+), no (%)	4 (3.6%)	13 (3.9%)	12 (3.6%)	0.976
Chronic renal disease (+), no (%)	4 (3.6%)	13 (3.9%)	12 (3.6%)	0.976
Chronic hepatic disease ^†^ (+), no (%),	7 (6.4%)	13 (3.9%)	14 (4.2%)	0.552
Autoimmune disease (+), no (%)	3 (2.7%)	6 (1.8%)	4 (1.2%)	0.549
Dementia (+), no (%)	21 (19.1%)	53 (16.1%)	59 (17.9%)	0.712
Baseline hemogram				
Hemoglobin, g/dL ± SD	12.1 ± 2.4	12.5 ± 2.0	12.5 ± 2.1	0.266
Absolute lymphocyte counts, /mm^3^, mean ± SD	376.0 ± 106.4	787.3 ± 136.7	1627.6 ± 742.5	<0.001
White blood cell counts, /mm^3^, mean ± SD	5995 ± 3990	5846 ± 3211	6322 ± 3876	0.232
Platelet, /mm^3^, mean ± SD	184,946 ± 98,719	186,860 ± 70,172	199,632 ± 74,163	0.058

Abbreviation: SD, standard deviation; ^#^ For a patient whose cancer had been completely cured was not accounted for in the analysis; ^$^ Comorbidities such as asthma or chronic obstructive lung disease were included; ^†^ Subjects with chronic hepatitis were included.

**Table 3 cancers-13-00471-t003:** Comparison of outcomes of COVID-19 among the three groups.

Outcomes	Absolute Lymphocyte Counts (Total *n* = 770)			*p*-Value
	Group I, <500/mm^3^ (*n* = 110)	Group II,≥500–<1000/mm^3^ (*n* = 330)	Group III ≥1000/mm^3^ (*n* = 330)	
Death, no (%)	44 (40%)	75 (22.7%)	45 (13.0%)	<0.001
Requirement of invasive ventilation, no (%)	45 (40.9%)	80 (24.5%)	49 (14.8%)	<0.001
Requirement of intensive oxygen supplements ^#^, no (%)	52 (47.3%)	105 (31.8%)	60 (18.2%)	<0.001
Requirement of oxygen supplements, no (%)	80 (72.7%)	183 (55.5%)	113 (34.2%)	<0.001
Total hospitalized period, days, mean ± SD	23.4 ± 13.9	25.5 ± 13.4	26.3 ± 11.8	0.120
Survival periods since admission for patients who died, days, mean ± SD	14.5 ± 13.3	16.4 ± 13.6	15.6 ± 14.4	0.764
Hospitalized periods for alive patient, days, mean ± SD	29.4 ± 10.8	28.2 ± 12.1	28.0 ± 10.5	0.631

Abbreviation: SD, standard deviation; ^#^ Intensive oxygen supply was accounted for when the subject used a facial mask, non-invasive ventilation, or high flow oxygen therapy for supplements of oxygen.

**Table 4 cancers-13-00471-t004:** Multivariate analysis of factors associated with the event of death.

Variable	Odds Ratio (95% CI)	*p*-Value
**Age**		
<60 years	1	
≥60 years	4.19 (2.14 -8.82)	<0.001
**Systolic blood pressure, baseline**		
<140 mmHg, no (%)	1	
≥140 mmHg, no (%)	1.66 (1.06 -2.6)	0.028
**Heart rate, baseline**		
<110/min	1	
≥110/min	2.34 (1.19–4.57)	0.013
**Dyspnea at presentation**		
Not present	1	
Present	4.18 (2.61 -6.81)	<0.001
**Mental disturbance at presentation**		
Not present	1	
Present	11.09 (3.28–47.25)	<0.001
**Diarrhea at presentation**		
Not present	1	
Present	0.60 (0.28–1.21)	0.171
**Comorbidity**		
Treating cancer, no	1	
Treating cancer, yes	3.15 (1.43 -6.85)	0.004
Diabetes, no	1	
Diabetes, yes	2.00 (1.24–3.21)	0.004
Hypertension, no	1	
Hypertension, yes	1.12 (0.69–1.8)	0.647
Chronic cardiac disease^#^, no	1	
Chronic cardiac disease^#^, yes	1.10 (0.57–2.08)	0.773
Chronic pulmonary disease^#^, no	1	
Chronic pulmonary disease^#^, yes	1.84 (0.68–4.74)	0.214
Chronic renal disease, no	1	
Chronic renal disease, yes	3.36 (1.32–8.54)	0.010
Dementia, no	1	
Dementia, yes	6.55 (3.84–11.40)	<0.001
**Hemoglobin**		
≥12.5 g/dL	1	
<12.5 g/dL	1.74 (1.10–2.77)	0.019
**Absolute lymphocyte counts**		
Group III, ≥1000/mm^3^	1	
Group II, ≥500–<1000/mm^3^	2.47 (1.50–4.13)	<0.001
Group I, <500/mm^3^	5.63 (3.0–10.72)	<0.001
**Platelet counts**		
≥100,000/mm^3^	1	
<100,000/mm^3^	2.34 (1.07–5.01)	0.031

## Data Availability

Restrictions apply to the availability of these data. Data was obtained from (Korea Disease Control & Prevention Agency) and are available (from the “is.cdc.go.kr”) with the permission of (Korea Disease Control & Prevention Agency).

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
