# Peer review of "Lymphopenia as a Biological Predictor of Outcomes in COVID-19 Patients: A Nationwide Cohort Study"

_cancers, 2021, doi:10.3390/cancers13030471_

Round 1

Reviewer 1 Report

In this study the authors using Korean nationwide cohort discussed the possibility to use lymphopenia as a prognostic marker for COVID-19 in the contest to control the transmission of virus.

Overall, this work is well structured. The authors have collected the data and by a statistical analysis explained how lymphopenia could help for intervention and treatment of COVID-19 patients.

I appreciated the discussion of several limitations and for this reason I suggest answering these points to state that lymphopenia can be a biological predictor. Moreover, it could be interest to know why there is no difference between three groups of ALC in terms of particular comorbidity (e.g. cancer with active treatment and autoimmune disease).

Author Response

Dear Editor and Reviewer 1,

My co-authors and I are more than pleased to submit the “revised version” of our manuscript titled “Lymphopenia as a biological predictor of outcomes in COVID-19 patients: A Nationwide Cohort Study” to be considered for publication in your journal.

First of all, thank you very much for your comprehensive review and insightful comments. We have revised the manuscript in accordance with the editor’s and reviewer’s suggestions. The responses to the reviewers are appended point-to-point.

Thank you for considering this manuscript for publication in your journal. I strongly feel that we were able to substantially improve our manuscript with your help.

I hope the revised manuscript will better meet the requirements of “Cancers” for publication.

Sincerely yours,

Dong-Gun Lee, MD, PhD and Dong-Wook Kim, MD and PhD. (Corresponding Authors)

Reviewer 2 Report

In this work, authors correlated the levels of lymphopenia with COVID-19 “aggressiveness” demonstrating how the severity of lymphopenia correlates with a more unfavorable prognosis. This allows the inclusion of lymphopenia levels among the prognostic markers for COVID-19. Despite the work sounds interesting, it gives the impression of being a little poor. I think it takes more data to render the work valuable for publication.

In particular:

  1. It is not clear to me if lymphopenia is such because it was caused by the virus itself, or if it was already present and was caused by other pathologies before the SARS-CoV2 infection or, again, the lymphopenia was caused by drugs that the patient was taking for other diseases. Please, explain thi point.
  2. The flow chart in figure 3 finally clarifies the workflow and how the patients have been chosen: I would suggest moving it to the beginning of the paper so that the reader is immediately clear on how this work was structured.
  3. In the work there are interesting ideas that are not dealt with in depth in the discussion: - which approaches causing lymphopenia should be avoided during the pandemic? (give some examples) - If lymphopenia is a marker of poor prognosis, how is it suggested to proceed with these patients who are the most at risk?
  4. I think that the evaluation of the lymphocyte count alone gives an indication, albeit precise, not complete. I would suggest analyzing individual lymphocyte subpopulations, such as B lymphocytes, T lymphocytes (CD4 and CD8) and NK cells.
  5. Furthermore, to give a more complete picture, I would suggest correlating overall survival even in patients without lymphopenia.
  6. In table 2, it is the percentage that gives a clear idea of the trend and not the absolute number. I would suggest emphasizing this data (for example put it in bold).

I appreciated that the authors included the study's weaknesses and limitations in their paper.

Author Response

Dear Editor and Reviewer 2,

We are pleased to submit the revised version of our manuscript titled “Lymphopenia as a biological predictor of outcomes in COVID-19 patients: A Nationwide Cohort Study” for consideration for publication in your journal.

We thank you for your comprehensive review and educative comments. We have revised the manuscript in accordance with the Editor’s and Reviewers’ suggestions and addressed the Reviewers’ comments point-by-point. Our responses to the Reviewers’ comments are appended in this document.

Thank you for considering this manuscript for publication in your journal. We believe that applying your recommendations has greatly improved our paper.

We hope the revised manuscript will better meet the requirements of Cancers for publication.

Sincerely,

Dong-Gun Lee, MD, PhD and Dong-Wook Kim, MD and PhD (Corresponding Authors)

Round 2

Reviewer 1 Report

After revision the pap is ok.

Reviewer 2 Report

I think the authors responded very well to the comments/criticisms. The manuscript, in its revised form, should be accepted for publication in Cancers.